# Metformin and the Liver: Unlocking the Full Therapeutic Potential

**DOI:** 10.3390/metabo14040186

**Published:** 2024-03-25

**Authors:** Federica Perazza, Laura Leoni, Santo Colosimo, Alessandra Musio, Giulia Bocedi, Michela D’Avino, Giulio Agnelli, Alba Nicastri, Chiara Rossetti, Federica Sacilotto, Giulio Marchesini, Maria Letizia Petroni, Federico Ravaioli

**Affiliations:** 1Department of Medical and Surgical Sciences, IRCCS Azienda Ospedaliero, Universitaria di Bologna, 40138 Bologna, Italy; federica.perazza@studio.unibo.it (F.P.); laura.leoni4@studio.unibo.it (L.L.); giulio.agnelli@studio.unibo.it (G.A.); alba.nicastri@studio.unibo.it (A.N.); chiara.rossetti7@studio.unibo.it (C.R.); federica.sacilotto@studio.unibo.it (F.S.); giulio.marchesini@unibo.it (G.M.); marialetizia.petroni@unibo.it (M.L.P.); 2Doctorate School of Nutrition Science, University of Milan, 20122 Milan, Italy; santo.colosimo@studio.unibo.it; 3Centro Hercolani S.r.l., 40123 Bologna, Italy; dr.alessandra.musio@gmail.com; 4U.O. Diabetologia, Ospedale C. Magati, Scandiano, 42019 Reggio Emilia, Italy; giulia.bocedi@ausl.re.it; 5S.C. Endocrinologia Arcispedale Santa Maria Nuova, 42123 Reggio Emilia, Italy; michela.davino@ausl.re.it; 6Division of Hepatobiliary and Immunoallergic Diseases, Department of Internal Medicine, IRCCS Azienda Ospedaliero, Universitaria di Bologna, 40138 Bologna, Italy

**Keywords:** NAFLD, NASH, MASLD, chronic liver disease, hyperglycaemia, type 2 diabetes mellitus, HCC, insulin resistance, metabolic syndrome, cirrhosis, nutrition, diet

## Abstract

Metformin is a highly effective medication for managing type 2 diabetes mellitus. Recent studies have shown that it has significant therapeutic benefits in various organ systems, particularly the liver. Although the effects of metformin on metabolic dysfunction-associated steatotic liver disease and metabolic dysfunction-associated steatohepatitis are still being debated, it has positive effects on cirrhosis and anti-tumoral properties, which can help prevent the development of hepatocellular carcinoma. Furthermore, it has been proven to improve insulin resistance and dyslipidaemia, commonly associated with liver diseases. While more studies are needed to fully determine the safety and effectiveness of metformin use in liver diseases, the results are highly promising. Indeed, metformin has a terrific potential for extending its full therapeutic properties beyond its traditional use in managing diabetes.

## 1. Introduction

Metabolic dysfunction-associated steatotic liver disease (MASLD), previously known as non-alcoholic fatty liver disease (NAFLD) [1], is a common condition that affects both adults and children, which is characterised by macrovesicular steatosis in ≥5% of hepatocytes, with no hepatocellular damage; metabolic dysfunction-associated steatohepatitis (MASH), also known as non-alcoholic steatohepatitis (NASH), is an advanced form of MASLD, identified by the presence of steatosis, hepatocyte ballooning, lobular inflammation, and fibrosis at histology [2]. People diagnosed with MASH experience a more severe progression of the disease, leading to advanced chronic liver disease (ACLD) and hepatocellular carcinoma (HCC) [3]. It is estimated that approximately 25% of the adult population and 3–10% of children are diagnosed with MASLD, with the number increasing up to 40% in children and to 70% in adults who are obese [4,5].

At present, no pharmacological therapies are approved for MASH, and current management approaches are focused on lifestyle modification, like a low-calorie diet, physical activity—both aerobic and resistance training—and cognitive behavioural therapy [6], to obtain weight loss and, at the same time, improve other related conditions, namely, blood pressure, triglycerides levels and insulin resistance. Diet is a cornerstone of MASLD management, primarily representing the first line of therapy. All international guidelines advise lifestyle modifications as a cornerstone of MASLD treatment, along with reducing alcohol intake; in addition, caloric restriction, generating a negative energy balance, has been associated with MASLD improvement [7]. A sustained weight loss of 10% has been proven to reduce liver fibrosis in patients with MASH [8,9].

Besides weight control, liver fibrosis benefits from enhanced insulin sensitivity and reduced glyco- and lipo-toxicity, which is also the aim of the therapy in Type 2 Diabetes Mellitus (T2DM), owing to the glucose-lowering action of anti-diabetic drugs [10,11,12,13].

Metformin is an oral anti-diabetic drug that improves insulin sensitivity and reduces glucose production in the liver. Its mechanisms of action extend beyond glycaemic control, making it a potential candidate for MASLD/MASH treatment. Several studies have investigated the effects of metformin on MASLD/MASH on liver histology and metabolic parameters in patients with MASLD with controversial results [8,14].

This review aims to provide a comprehensive overview of the current knowledge and research, consolidate the existing evidence, and stimulate further scientific inquiry to unlock the full therapeutic potential of metformin and lifestyle interventions in liver-related conditions beyond its traditional role in managing T2DM.

## 2. Metformin’s Mechanism of Action and Role in Liver Disease

### 2.1. Historical Backgrounds and Absorption

Metformin has significant benefits concerning glucose metabolism and diabetes-related complications. The chemical structure of metformin is derived from galegine, a natural product extracted from the plant Galega officinalis [15]. Chemical analyses of Galega officinalis dating from the mid-1800s discovered the plant to be abundant in guanidine, which was reported to lower animal blood glucose [16]. Although metformin shares structural similarities with glucose-reducing mono- and diguanidines, its blood glucose-lowering effects in animals (rabbits and dogs) were only reported in 1929 [17,18]. In 1957, Jean Stern converted the blood glucose-lowering potential of metformin in therapy for T2DM: he observed that, in subjects with maturity-onset diabetes, metformin could replace or reduce the need for insulin without the occurrence of frank hypoglycemia [19,20,21,22,23]. Currently, metformin is still considered among the first-line treatments for T2DM [24].

Approximately 70% of the metformin dose is absorbed from the small intestine, while the remaining part passes into the colon before being excreted in faeces. Metformin is excreted in the urine unchanged, with no metabolites reported [25].

Metformin uptake requires organic cation transporter 1 (OCT1), which is highly expressed in the liver, kidneys, and small intestines, with little uptake from the peripheral tissues. OCT3 and multidrug and toxin extrusion 1 (MATE1) transporters play a minor role in metformin uptake [25]. Metformin uptake is saturable and dose-dependent, consisting of a predominantly transporter-dependent mechanism; accordingly, genetic variations in the transporters or transporter-inhibiting drugs could affect its uptake and tolerance [26]. Modified-release formulations have been developed to enhance gastrointestinal tolerability, spreading the absorption of metformin along the gut and reducing local concentrations [27,28].

### 2.2. Mechanism of Action of Metformin

The most important mechanism of action of metformin occurs in the liver, where metformin regulates hepatic glucose production. Other benefits have been linked to alterations in the composition of the gut microbiome, intestinal glucose uptake, and hormone secretion (e.g., growth differentiation factor 15 [GDF15], glucagon-like peptide-1 [GLP-1]).

#### 2.2.1. Regulation of Hepatic Gluconeogenesis

Hepatic glucose production depends on hepatic gluconeogenesis, glycogenolysis, glycogen synthesis, and glycolysis. These biochemical mechanisms lead to 85% to 90% of endogenous glucose production [29].

The glucose-lowering effect of metformin is primarily due to its role in regulating hepatic gluconeogenesis. Metformin causes a modest inhibition of mitochondrial respiratory chain Complex I, leading to a mild reduction in adenosine triphosphate (ATP) synthesis and a concomitant increase in adenosine monophosphate (AMP) cellular levels. The metformin-induced reduction in hepatic gluconeogenesis is an ATP-dependent process that could result from decreased ATP levels [30]. Moreover, increased AMP levels result in the inhibition of the activity of enzymes that are regulated by AMP and are involved in gluconeogenesis, such as adenylate cyclase and fructose-1-6-bisphosphatase (FBP1), which contributes to reducing glucose output [31].

In addition, in Complex I, metformin directly down-regulates mitochondrial glycerol 3 phosphate dehydrogenase (mGPDH), leading to an increased cytosolic redox state—with a reduction in nicotinamide adenine dinucleotide reduced form (NADH) and increased nicotinamide adenine dinucleotide (NAD)—to reduced gluconeogenesis from lactate, and to a decrease in the activity of the glycerol–phosphate shuttle (which transfers NADH from the cytosol to mitochondria). Furthermore, metformin elevates the hepatic redox state by increasing the glutathione to oxidised glutathione ratio (GSH:GSSG), inhibiting genes encoding the enzymes implicated in gluconeogenesis. Thus, metformin inhibits mitochondrial respiratory chain complex IV, indirectly inhibiting mGPDH activity [26].

Furthermore, metformin activates AMPK, which phosphorylates nuclear receptor TR4 to prevent its transcriptional activation, inhibiting the expression of the TR4-mediated gene for coenzyme A dehydrogenase-1 [32]. AMPK also phosphorylates SREBP-1c, which inhibits proteolytic maturation, nuclear translocation of SREBP-1c, and the expression of the downstream fatty acid synthase gene, thus increasing fatty acid β-oxidation and reducing the de novo synthesis of triglycerides (TG) [33] and their liver accumulation, finally preventing hepatocyte steatosis [34] (Figure 1).

#### 2.2.2. Metformin and Glucagon-Like Peptide 1

Metformin treatment has been shown to increase the concentration of GLP-1, a hormone derived from the proglucagon gene. GLP-1 regulates blood glucose levels by enhancing insulin secretion, inhibiting glucagon secretion, stimulating the growth of beta cells, delaying stomach emptying, and promoting a sense of satiety and fullness [35]. Thus, GLP-1 has an anorexigenic activity, reducing both homeostatic and hedonic feeding due to a loss of interest in food and a decreased appetite [36,37,38]. Being a potent insulin secretagogue, GLP-1 has a brief half-life in the body due to rapid cleavage by the widely expressed enzyme DPP-4 (dipeptidyl peptidase-4), whose levels are reduced during metformin treatment [39]. In conclusion, metformin treatment increases the levels of GLP-1 and reduces those of DPP-4, enhancing GLP-1 properties in blood glucose and appetite management.

#### 2.2.3. Metformin and Skeletal Muscle

Glucose uptake in skeletal muscle is mediated by insulin signals, which determine the translocation of glucose transporter 4 (GLUT4) from the cytosol to the plasma membrane [40]. GLUT4 is also expressed in adipose tissue and cardiac muscle; its dysfunction in adipose tissues and skeletal muscle is associated with glucose uptake and lipid synthesis in the liver [41]. Skeletal muscle is responsible for the majority of insulin-stimulated glucose disposal in the body [42,43], and it is involved in insulin resistance, a condition in which peripheral tissues lose the ability to uptake glucose from the bloodstream; thus, muscle has a crucial role in the development and progression of metabolic diseases such as T2DM and obesity [44,45,46]. Metformin activates AMPK in skeletal muscle, leading to increased translocation of GLUT4 to the cell membrane and, thereby, increased glucose uptake. This action reduces insulin resistance and liver steatosis [16].

#### 2.2.4. Gut Glucose Uptake and Intestinal Microbiota Modification

Recently, it has been reported that metformin impacts glucose uptake from the intestinal tract [28,47]. A study by Ito, conducted using positron emission tomography (PET), has shown that metformin administration is associated with the intestinal accumulation of i.v.-injected [18F] fluorodeoxyglucose (FDG), a non-metabolizable glucose derivative, thus promoting glucose transport from the circulation into excrement [48]. Moreover, metformin alters the gut microbiome’s composition, contributing to its therapeutic effects [49]. *Akkermansia muciniphila* is reduced in patients with prediabetes (impaired glucose tolerance and/or impaired fasting glucose) and with a newly detected T2DM, Indicating that a low abundance of this bacterium might be a biomarker for glucose intolerance [50]. Metformin treatment is associated with the increased presence of *Akkermansia muciniphila* in the gut lumen, which promotes the maintenance of intestinal barrier integrity, the increment of short-chain fatty acids with positive actions on peripheral tissues (adipose tissue, skeletal muscle, and liver) by improving insulin sensitivity [49,51,52], and reduced reabsorption of bile acids in the distal ileum, resulting in increasing bile salt levels within the colon, which might modify its microbiota [53,54] (Figure 1).

#### 2.2.5. Metformin and Peptide Hormone Growth/Differentiation Factor 15 (GDF15)

GDF15, a member of the transforming growth factor β superfamily, is expressed in various tissues, primarily the liver and the kidney, but also in white and brown adipose tissues and skeletal muscle, and plays an important role in the integrated cellular stress response [55]. High levels of GDF15 are associated with reduced food intake and weight loss [56]; these actions are largely centrally mediated because GDF15 binds to its receptor, the glial cell line-derived neurotrophic factor family receptor alpha-like (GFRAL), which is localised in the hindbrain and hypothalamus [57,58,59]. GDF15-induced weight loss in rodents is accompanied by increased insulin sensitivity and improved glucose tolerance [57,58,59,60,61]. Furthermore, in subjects with obesity, bariatric surgery was shown to increase the plasma levels of GDF15 and insulin sensitivity, suggesting a possible positive role of GDF15 on insulin activity in humans [62]. Sjøberg et al. recently proved that treatment with GDF15 ameliorated insulin sensitivity independently of weight loss in mice and rats [63]. Finally, metformin increases circulating levels of GDF15, which explains the role of metformin in energy balance and body weight control [64,65,66,67] (Figure 1). In conclusion, GDF15 may represent a future target therapy for T2DM and obesity.

#### 2.2.6. Metformin and Platelet-Derived Growth Factor (PDGF)

The PDGF signalling pathway is one of the pathways used in activating hepatic stellate cells (HSC) [68]. PDGF-BB, a subtype of the PDGF family, is induced by the most potent stimulator of HSC growth and intracellular signal transduction [68]. PDGF activates extracellular signal-regulated kinase and the protein kinase B (Akt)/mammalian target of rapamycin (mTOR) pathways, which are serine/threonine protein kinases which are crucial in cell growth, differentiation, proliferation, migration, and survival [68]. Metformin activates AMPK to regulate PDGF-BB-induced phenotypic changes in HSC activation [69,70]. Adachi et al. demonstrated that metformin inhibits PDGF-induced phosphorylations, resulting in the inhibition of HSC proliferation and migration and a reduction in extracellular matrix secretions consisting of type I collagen and fibronectin, which leads to the inhibition of fibrosis [69,71].

#### 2.2.7. Metformin and Mitochondria

Mitochondria are crucial in the energetic homeostasis of cells. Subjects affected by T2DM have a reduction in number of mitochondria and their respiratory activity in the liver and other metabolic tissues, and mitochondrial dysfunction is implicated in the development of T2DM [72].

Human studies discovered that metformin promoted mitochondrial respiratory chain activity in various tissues [73,74]; in this context, it has been found that metformin significantly increased mitochondrial complex 1 activity in the livers of mice [75,76]. Consequently, pharmacological metformin concentrations increased mitochondrial oxidative phosphorylation in liver cells [76,77] and promoted mitochondrial fission through the phosphorylation of mitochondrial fission factor and the recruitment of dynamin-related protein 1 (DRP1) by AMPK [76,78]. In liver-specific Drp1 knockout mice, a marked reduction in mitochondrial respiration was observed, associated with increased lipid accumulation [76]. Since mitochondrial fission is connected with oxidative phosphorylation, metformin-mediated mitochondrial fission increases nutrient oxidation in mitochondria. In addition, metformin-promoted fission eliminates compromised mitochondria via mitophagy to maintain a healthy mitochondrial population [79].

#### 2.2.8. Regulation of Lipid Metabolism

Metformin also has a role in the regulation of lipid metabolism. Indeed, in patients with T2DM or insulin resistance, high insulin levels are related to the dysregulation of intestinal lipoprotein metabolism [80,81,82,83]. The expression of lipogenic genes involved in de novo lipogenesis is controlled by the sterol regulatory element-binding protein-1c (SREBP-1c) expressed in the jejunum and ileum [84]. SREBP-1c is positively regulated by insulin and negatively by AMPK and promotes enzymes, i.e., acetyl-CoA carboxylase (ACC1) and fatty acid synthase (FAS), which are involved in de novo fatty acid synthesis [85,86]. Metformin can reduce intestine-derived triglyceride-rich lipoproteins measured in the plasma (−50% chylomicrons and −20% chylomicron remnant lipoprotein fractions) of T2DM patients [83]. Moreover, metformin induced a small decrease in mRNA expression of SREBP-1c and ACC1, causing a moderate amelioration of intestinal lipid homeostasis [82]. In addition, a reduced chylomicron concentration could also be determined by an increased GLP-1 concentration in the intestine, leading to a reduction in apo B-48, triglyceride availability, and chylomicron secretion [87,88].

Furthermore, in the liver, the activation of AMPK provided by metformin is connected to the modulation of cholesterol synthesis, as the phosphorylation of 3-hydroxy-3-methyl-glutaryl-coenzyme A reductase (HMGCR) leads to a reduction in cholesterol biosynthesis [89]. Other than the phosphorylation of AMPK, in hepatoma cells of rats, downregulated mRNA levels of HMGCR and HMG-CoA synthase (HMGCS) were identified; this phenomenon was detected with high doses of metformin [90], but it was not registered with lower doses. The inhibition of HMGCR was associated with decreased triglycerides and LDL levels in plasma [91]. Unfortunately, a minimal reduction in HMGCR activity in human-cultured fibroblasts was detected with metformin treatment [92]. On the one hand, in 1983, Scott et al. demonstrated that hepatic HMGCR was not affected by metformin treatment; on the other hand, intestinal HMGCR showed a reduction in activity of 62%. The Acyl-CoA cholesterol acyltransferase (ACAT) in the catalysis of cholesterol esters also showed decreased activity (−35%) [93]. In this context, the action of metformin in regulating lipid metabolism is mainly explained by its activity in the intestine. Accordingly, inhibiting the intestinal absorption of bile acids mediated by metformin determined an increased synthesis of bile acids in the liver, and cholesterol is used for this process, leading to a reduction in cholesterol inside the hepatocytes [94,95]. The up-regulation of the LDL receptor may increase the uptake of lipoproteins, restoring an adequate cholesterol concentration in the liver. Hereby, metformin may indirectly decrease LDL and total plasma cholesterol concentrations [96].

## 3. Metformin in Clinical Practice

### 3.1. Efficacy of Metformin in Liver Diseases

#### 3.1.1. Metabolic Dysfunction-Associated Steatotic Liver Disease (MASLD)

Preclinical studies have shown that metformin has beneficial effects in animal models of MASLD, reducing hepatic steatosis, inflammation, and fibrosis [97]. Treatment with metformin might reverse fatty liver disease in obese, leptin-deficient mice by inhibiting the hepatic expression of tumour necrosis factor (TNF)-α and TNF-inducible factors that promote hepatic lipid accumulation and ATP depletion [98]. Despite promising data on animal models, clinical studies evaluating the efficacy of metformin in MASLD have yielded mixed results.

Clinical studies demonstrated that metformin significantly reduces liver enzymes, improving aspartate and alanine amino-transferase (AST and ALT) levels [99,100,101,102,103,104,105,106,107,108,109,110]. Three studies also testified to improvements in MASLD after metformin treatment in 2010, 2013 and 2019, in which a reduction in intrahepatic fat content was assessed by ultrasonography [111,112,113]. Furthermore, metformin improved the amount of fat measured by controlled attenuation parameter (CAP, performed with FibroScan™) after three and five months of therapy [114]. Several meta-analyses confirmed that metformin ameliorates MASLD, improving ALT and AST levels [115,116,117]. These data have also been confirmed by a recent meta-analysis published in 2022, in which metformin decreased circulating ALT and AST levels [115].

On the other hand, some studies did not show significant improvements in ALT and AST levels after metformin treatment [118,119,120,121]. Similarly, in a trial published in 2023, metformin was demonstrated to be less effective in reducing GGT levels and fat content measured with CAP in patients with a new diagnosis of T2DM [122]. In a randomised, double-blinded, placebo-controlled clinical trial conducted on 173 patients (8–17 years) with biopsy-confirmed MASLD, metformin was not superior to the placebo in achieving a sustained reduction in ALT levels [123]. In this context, in two systematic reviews and meta-analyses, metformin did not ameliorate ALT levels [4,124]. Moreover, hepatic fat content, evaluated by ultrasonography, computed tomography, and proton magnetic resonance spectroscopy, did not improve after 16–48 weeks of metformin therapy compared with the baseline [125]. Finally, in a meta-analysis including four randomised controlled trials (RCTs) with 309 paediatric patients with MASLD, treatment with metformin failed to improve liver enzymes statistically, but was beneficial in enhancing lipid parameters and insulin metabolism [126].

In conclusion, the heterogeneity across study designs, patient cohorts, and treatment regimens may underlie the observed outcome disparities. Nevertheless, we also suggest what was emphasised in the meta-analysis mentioned, demonstrating that metformin could be a potential modality for reducing liver enzyme elevation and decreasing hepatic fat content.

#### 3.1.2. Metabolic Dysfunction-Associated Steatohepatitis (MASH) and Fibrotic-MASH

Currently, there is no standardised treatment for MASH, and drugs that counteract insulin resistance, necro-inflammatory activity, and fibrosis have been used to reduce inflammation and fibrosis.

In 2004, Nair et al. conducted a pilot trial and demonstrated that one year of therapy with metformin improved the histological features of liver steatohepatitis (steatosis, inflammation and fibrosis), which was proven by biopsy [109]. Loomba et al. and Torres et al. later confirmed this data in two trials [101,106]. In 2019, an open-label randomised trial conducted for 48 weeks, including patients with biopsy-proven MASH, demonstrated that the association between N-acetylcysteine and metformin could reduce liver disease activity in patients with MASH, testified by the improvement in ballooning and NAS [127], and in a randomised, double-blind, placebo-controlled trial conducted on 173 children affected by MASLD, metformin (−0.3 vs. 0.1) and vitamin E (−0.5 vs. 0.1) significantly improved the ballooning degeneration score compared with the placebo [123]. In a retrospective study including 1292 patients with T2DM, 48 out of 83 patients (57.8%) who had advanced fibrosis (defined by Fibrosis-4 index > 2.67) experienced a regression of fibrosis after two years of treatment with metformin. Unfortunately, this study lacks a control group and uses a surrogate marker for liver fibrosis [128]. However, in 2009, in a randomised controlled trial, it was demonstrated that combined therapy with metformin and pentoxifylline vs. a prescriptive diet was not associated with a statistical difference between the two groups in terms of histological parameters, although there was a tendency for NAS to be reduced in the metformin plus pentoxifylline group [129].

In a group of 166 paediatric patients affected by MASLD/MASH, Gawrieh et al. noticed that ELF (enhanced liver fibrosis score), useful for detecting different stages of fibrosis, did not show a significant decrease with fibrosis improvement since neither vitamin E nor metformin resulted in substantial improvement in fibrosis [130]. In a randomised open-label study involving 64 patients with T2DM or impaired glucose tolerance, no substantial improvement in histological parameters was detected by liver biopsy, even though only ten patients out of twenty-two patients were biopsied and treated with metformin [105].

On the other hand, in an open-label, randomised trial, including 55 non-diabetic patients who received metformin and 12 patients and 28 controls who received vitamin E or a weight-reducing diet, the histological assessment showed a significant improvement in fibrosis and necro-inflammation when compared to baseline. Unfortunately, a liver biopsy was only performed in the metformin group [104].

Many systematic reviews have been conducted, and globally, it has been concluded that there is insufficient evidence to support metformin’s efficacy in treating MASH [124,131]. Even in a previous meta-analysis of nine randomised controlled trials, it was found that metformin improved liver histologic scores for steatosis and ballooning without any significant improvement in fibrosis and lobular inflammation [132]. Recently, Mantovani et al. [133] evaluated the use of metformin (1000–2000 mg/die) in six RCTs (573 individuals treated for a median of 9 months; 5/1 study on adults/children) and concluded that, except for the paediatric trial, metformin showed a small beneficial effect on histologic steatosis and inflammation but not on liver fibrosis. When evaluated by imaging, the impact of metformin on steatosis was neutral, as it was on BMI, whereas it was associated with a significant reduction in serum aminotransferase levels (especially serum ALT) and an improvement in HbA1c levels [133].

In summary, although several studies tested metformin for MASH treatment, not all are methodologically sound. The actual evidence shows that the use of metformin does not lead to any significant improvement in histological parameters such as steatosis, ballooning, and lobular inflammation. Therefore, further research and investigations are required to fully understand its potential benefits for MASH improvement.

#### 3.1.3. Advanced Chronic Liver Disease (Cirrhosis) and Complications

Advanced chronic liver disease (ACLD) is the end-stage of chronic liver disease characterised by extensive fibrosis and the disruption of normal liver architecture. ACLD is associated with numerous complications, such as portal hypertension, ascites, hepatic encephalopathy, and an increased risk of hepatocellular carcinoma (HCC). Metformin has been shown to have potential effects on liver fibrosis and disease progression [134], but the exact mechanisms are not fully understood. Insulin resistance is commonly observed in ACLD-related MASLD and is associated with disease progression [68]. Metformin’s ability to improve insulin sensitivity may have implications for slowing down the progression of cirrhosis [135]. Additionally, the anti-inflammatory effects of metformin and its role in reducing oxidative stress could attenuate the inflammatory response and fibrogenesis seen in cirrhosis [136].

##### Portal Hypertension-Related Complications

The development of portal hypertension (PH) is a major inflexion point in the natural progression of ACLD, conferring the risk of hepatic decompensation with ascites, variceal bleeding, and other PH-related syndromes [137]. Mechanistically, PH is caused by increased intrahepatic vascular resistance due to the distortion of liver architecture by fibrosis, a dynamic increase in sinusoidal tone, and progressive splanchnic vasodilation [137]. PH and clinically significant portal hypertension (CSPH) are defined by hepatic venous pressure gradients (HVPG) of ≥5 mmHg and ≥10 mmHg, respectively [138]. In a trial conducted by Triphati et al., cirrhotic mice models were treated with metformin, resulting in a reduction in portal pressure [139]. Studies in rodents suggest that metformin increases hepatic nitric oxide bioavailability, potentially through AMPK-dependent mechanisms that ultimately reduce hepatic vascular resistance [139,140,141,142]. Also, in a murine model of cirrhosis, metformin was associated with reduced liver fibrosis and intrapulmonary shunts [143]. Based on these observations, in 2021, Ritting et al. observed that HVPG was acutely reduced after administering metformin to 16 cirrhotic patients, compared to placebo [144]. In conclusion, based on limited evidence, metformin improves HVPG and may reduce complications related to high portal pressure in patients with ACLD.

##### Hepatic Encephalopathy

Hepatic encephalopathy is a syndrome characterised by anxiety, cognitive, memory, and learning impairment, balance problems, and personality changes; it may eventually result in a coma and, ultimately, death [145]. This deterioration of brain function is due to the liver’s incapacity to remove blood toxins, such as ammonia and lipopolysaccharides, which causes systemic inflammation and activation of the circulatory neutrophils [146]. Then, ammonia and other toxic agents move to the brain, generating pathological changes such as neuroinflammation and neuropathy [147]. Because metformin contributes to intestinal barrier integrity and prevents bacteria translocation from the gut to the bloodstream, it may have a role in preventing hepatic encephalopathy [148].

##### Hepatocellular Carcinoma

Hepatocellular carcinoma (HCC) is the most common primary liver cancer and a major global health concern [149,150]. It is closely associated with ACLD. Metabolic disorders, such as obesity, T2DM, and insulin resistance, have emerged as significant risk factors for the development and progression of HCC [151].

Metformin, in addition to its role as an antidiabetic agent, has garnered attention for its potential anticancer properties in several neoplasia [152,153,154,155,156], although the mechanisms underlying its anticancer effects are not fully understood [26]. Preclinical studies using in vitro and in vivo models have provided valuable insights: metformin might inhibit cell proliferation [157] by inducing cell cycle arrest [158], and promoting apoptosis [159] through the AMP-activated protein kinase (AMPK) pathway [160], the mammalian target of rapamycin (mTOR) pathway [157], and the insulin-like growth factor (IGF) signalling pathway [161]. Metformin has also been found to inhibit angiogenesis [162], reduce cancer stem cell properties, and enhance the efficacy of conventional anticancer therapies [154]. Metformin also has indirect antitumoral effects due to its ability to reduce insulin and insulin-like growth factor 1 (IGF-1) levels and to influence metabolic pathways such as lipid metabolism and inflammation [15], which are known to promote HCC tumour growth and proliferation [162,163,164,165].

In this context, Kaplan et al. demonstrated a null association between metformin therapy and the occurrence of HCC [166]. On the other hand, concerning the risk of HCC development, You et al. reported metformin to be potentially chemopreventive [167]. In 2017, a meta-analysis of nineteen studies (two RCTs, ten cohorts, seven case-control studies) involving patients with T2DM showed that metformin reduces the risk of HCC by 48% in patients treated with metformin (OR = 0.52; 95% CI: 0.40–0.68), compared with non-users [168]. Later, Li et al. also confirmed the association between metformin and a decreased risk of HCC in subjects with T2DM [169], but a 2023 meta-analysis by Zeng et al. failed to reach statistical significance (HR: 0.57; 95% CI: 0.31–1.06) [170].

Metformin has also been associated with increased survival in patients with HCC, as shown by a meta-analysis including six retrospective cohort studies [171]. Treatment with metformin was associated with longer overall survival at one (OR = 2.62, 95% CI: 1.76–3.90), three (OR = 3.14, 95%CI: 2.33–4.24), and five years (OR = 3.31, 95%CI: 2.39–4.59), and with significantly longer recurrence-free survival at one (OR = 2.52, 95%CI: 1.84–3.44) and three years (OR = 2.87, 95%CI: 2.15–3.84) but not at five years (OR = 2.26, 95%CI: 0.94–5.45) after curative treatment including hepatic resection and radiofrequency ablation therapy. These data were confirmed in 2022 by a meta-analysis in which metformin utilisation after HCC curative treatment increased 3-year [OR = 1.50, 95% CI: 1.22–1.83] and 5-year (OR 1.88, 95% CI: 1.47–2.41) overall survival and decreased 1-year (OR = 1.31, 95% CI: 1.08–1.59), 3-year (OR = 1.88, 95% CI: 1.48–2.37), and 5-year (OR = 1.83, 95% CI: 1.40–2.40) recurrence rates [172]. On the other hand, metformin treatment was not associated with prolonged overall and recurrence-free survival after non-curative HCC treatment (systemic therapy with sorafenib) compared to insulin treatment [171].

Clinical studies are being conducted to investigate the potential of metformin in treating HCC. Recent studies have reported that patients with HCC treated with metformin showed improved overall and recurrence-free survival. Due to its immunomodulatory properties, metformin could be combined with other therapeutic strategies for HCC management. However, further studies are necessary to fully understand the potential of metformin treatment in HCC [173].

### 3.2. Potential Preventive Effects of Metformin in High-Risk Populations

Different studies have investigated the effect of metformin in populations at increased risk of MASLD. Metformin has also been associated with a reduction in the prevalence of metabolic syndrome and liver involvement in a study including 140 overweight patients with hyperinsulinemia and PCOS treated with metformin for 12 months [174]. Four meta-analyses, including studies on the use of metformin in MASLD, revealed substantial improvement in blood cholesterol levels, fasting plasma glucose, and haemoglobin A1c (HbA1c), suggesting that metformin can be helpful as a treatment against MASLD risk factors [8,124,175,176]. Huang et al. [115], in a meta-analysis with network pharmacology—a discipline which attempts to understand drug actions and interactions with multiple targets—including ten studies (both RCTs and non-RCTs) involving 576 patients with MASLD, highlighted an association between metformin and reduced triglyceride (TG) levels (mean decrease (MD) = −0.17, 95% CI = −0.26 to −0.08), total cholesterol (MD = −0.29, 95% CI = −0.47 to −0.10), and insulin resistance (MD = −0.42, 95% CI = −0.82 to −0.02). The authors found no association with reduced body mass index (BMI (MD = −0.65, 95% CI = −1.46 to 0.16). On the other hand, in another network, a meta-analysis by Huang et al. [177], including twenty-two RCTs involving 1377 patients, a poor performance was observed for metformin when compared to other antidiabetic drugs such as the GLP-1 receptor agonist for hepatic fat content, the MASLD activity score, ALT, AST, GGT, and body weight. In conclusion, despite its insulin-sensitizing activity, metformin has not been proven to play a role in MASLD prevention in clinical studies.

### 3.3. Drug-Induced Liver Injury (DILI) and Chemical-Induced Liver Injury

DILI refers to various types of liver injury during medication exposure, resulting from metabolite hepatotoxicity or poor drug tolerance in specific patient populations. The incidence is increasing yearly in China and Western countries due to substance abuse [178]. DILI is the most common cause of acute liver failure [179]; chemical hepatotoxicants determine chemical-induced liver injury and are similar to DILI in terms of mechanism.

In Western countries, acetaminophen (APAP) overdose is the most frequent cause of DILI and acute liver failure [180]; this event is related to excessive APAP consumption, which is converted to N-acetylbenzoquinone imine (NAPQI) by hepatic cytochrome P4502E1. It consumes reduced glutathione in mitochondria, and the remaining NAPQI then reacts to form covalent links with biological macromolecules, proteins in particular, resulting in mitochondrial damage and necrotic cell death. This phenomenon increases radical oxygen species (ROS) production, determining Jun N-terminal kinase (JNK) phosphorylation and activation, which contributes to liver cell death [181]. It has been found that metformin has a protective role against APAP overdose-provoked hepatotoxicity via growth arrest and DNA damage 45β (GADD45β)-dependent JNK regulation. Metformin can enhance the expression of growth arrest and GADD45β to inhibit the phosphorylation of mitogen-activated protein kinase 4, inhibiting JNK phosphorylation to protect hepatocytes from oxidative damage [182]. However, another study reported that metformin does not prevent JNK activation or mitochondrial JNK translocation but reduces APAP protein adducts in mitochondria. Moreover, metformin inhibits mitochondrial respiratory chain complex I, which reduces proton leak and ROS generation in liver cell mitochondria, thereby reducing hepatocyte apoptosis to treat DILI [183].

Metformin is also protective against chemical-induced liver injury. D-galactosamine increases Lipopolysaccharides (LPS) and Tumor Necrosis Factor α (TNF-α), determining liver injury. Metformin is able to decrease inflammatory indicators such as myeloperoxidase and malondialdehyde by promoting the classic AMPK signalling pathway to inhibit apoptosis induced by LPS and TNF-α, thus mitigating liver damage [184,185]. Furthermore, arsenic trioxide, a chemotherapy substance used to treat promyelocytic leukaemia, causes liver injury by generating ROS, while metformin can inhibit mitochondrial respiratory chain complex I and increase NAD+/NADH ratio, protecting against arsenic trioxide damage. Moreover, metformin can inhibit carbon tetrachloride-induced hepatotoxicity, possibly connected with the increase in glutathione in hepatocyte mitochondria [186].

## 4. Adverse Effects and Safety Profile of Metformin in Liver Disease Management

### 4.1. Common Side Effects of Metformin

Metformin tolerance is limited by the side effects experienced by many patients. The most common are gastrointestinal tract disorders such as diarrhoea, nausea, vomiting, flatulence, abdominal pain, and loss of appetite. Approximately 25–30% of patients report these effects, leading to treatment withdrawal in 5% of cases [28], although the modified-release formulation is more tolerable [187].

#### 4.1.1. Gastrointestinal Side Effects

Different pathophysiological hypotheses have been proposed to explain metformin-induced gastrointestinal effects. These theories include genetic variations in the transporter, stimulation of the intestinal secretion of serotonin, production of lactic acid by the intestinal mucosa, bile-salt malabsorption, and changes in the microbiome [188].

Among the pharmacokinetic mechanisms, genetic variations in the organic cation transporter-1 (OCT1) seem to be involved [189]. OCT1 is predominantly expressed in the liver but plays an important role in transferring metformin from the gut lumen to the interstitium. According to recent papers, this would take place on the apical surface of intestinal epithelial cells, with PMAT (plasma membrane monoamine transporter) and SERT (serotonin transporter) [190] being actively involved. In the GoDARTS study, individuals with two reduced-function OCT1 alleles who were treated with OCT1 inhibitors, such as tricyclic antidepressants, citalopram, proton-pump inhibitors, and spironolactone, were over four times more likely to develop intolerance [189]. OCT3, another member of the solute carrier family 22 (SCL22), is instead mainly expressed in the skeletal muscle but is also expressed in the intestine, and it is associated with metformin efflux in the salivary glands, leading to dysgeusia during metformin treatment [191].

Serotonin released by mucosal enterochromaffin cells stimulates 5-hydroxytryptamine receptors number 3 (5-HT3) and 5-HT4 to induce or augment peristalsis and propulsion [192]. Metformin may cause gastrointestinal side effects by inhibiting the SERT-mediated serotonin re-uptake due to increased gastrointestinal motility induced by serotonin [189,192].

Diarrhoea associated with metformin treatment could also result from the osmotic effect of increased luminal bile salt. Bile acid absorption in the ileum is an active process involving the nuclear farnesoid X receptor (FXR), which can be phosphorylated by AMPK, decreasing FXR transcription. Through AMPK activation, metformin could be responsible for increased bile salt presence in the intestinal lumen, leading to alterations in the microbiome and stool consistency [193].

Concerning the microbiome, adverse gastrointestinal effects could be mediated by the relative abundance of *Escherichia* spp. in people treated with metformin [194]. *Escherichia* spp. has been functionally associated with gas metabolism, and the prevalence of bloating and flatulence in people treated with metformin is around 25% and 8%, respectively [195]. *Akkermansia*’s presence increases with metformin treatment, contributing to maintaining intestinal barrier integrity, but it is unclear whether it plays a role in the development of gastrointestinal side effects (Figure 2).

#### 4.1.2. Metformin Associated Lactic Acidosis (MALA)

MALA is a rare condition resulting in altered lactate and hydrogen metabolism defined as pH < 7.35 and lactate > 5.0 mmol/L in the setting of metformin use or overdose [196]. A proposed mechanism suggests that metformin reduces mitochondrial complex activity, increasing enterocyte glycolysis and lactate production to maintain energy homeostasis [197]. In the incidental form of MALA, predisposing pathophysiological conditions determine supratherapeutic metformin levels in plasma; in particular, this situation occurs in the presence of chronic comorbidities such as kidney, liver, and heart failure, shock, and critical illness [196]. The risk is minimised when metformin is used appropriately, the dosage is adjusted according to kidney function, and contraindications are considered [198]. The exact mechanism by which metformin can contribute to lactic acidosis has yet to be fully understood [199]. Metformin inhibits mitochondrial complexes in the electron transport chain, leading to decreased aerobic metabolism and a subsequent increase in lactate production; if lactate clearance is impaired due to conditions such as kidney dysfunction, lactate accumulation in the blood may occur [200]. Moreover, although reduced gluconeogenesis is a desired effect in managing T2DM, it can potentially lead to lactic acidosis if glucose metabolism is compromised and lactate production increases [201]. In conclusion, lactic acidosis is not necessarily exclusively due to metformin accumulation, and the overall prognosis depends on the underlying conditions [202] (Figure 2); elevated alcoholic consumption is connected to an increased risk of MALA [203].

### 4.2. Safety of Metformin in Patients with Liver Disease

Despite the potential benefits of metformin, the FDA’s official ‘label’, under ‘Warnings and Precautions’, warned against its use in hepatic impairment due to the risk of lactic acidosis [204]. This advice derives from historical concerns that the reduced hepatic clearance of lactate in chronic liver disease may enhance the risk of lactic acidosis associated with metformin therapy [205,206], but the evidence supporting this warning is limited. In 1979, Woll reported that subjects with ACLD had a prolonged lactate half-life compared to a control group [206]; in 2012, Jeppesen et al. recruited 142 individuals with ACLD and 14 healthy controls and demonstrated that, after stimulation with galactose, lactic acid levels were more elevated in cirrhotic individuals than in the control group. Lactate levels seemed to increase with the severity of ACLD [205]. The mildly elevated plasma lactate levels in ACLD subjects vs. those without ACLD with lesser degrees of fibrosis may stem from a major decrease in hepatic blood flow and, thus, reduced hepatic uptake and elimination of lactate [205].

On the other side, in 2020, Smith and al., in a cross-sectional study evaluating the safety of metformin in patients affected by all-cause CLD, with or without T2DM [204], reported that metformin and lactate levels in plasma remained below the considered safety thresholds of 5 mg/L and 5 mmol/L, respectively. Considering that only lactate concentrations above 5 mmol/L are related to a major risk of lactic acidosis [25,206,207,208], when the dose of metformin is chosen to be in line with renal function, plasma metformin concentrations stay below 5 mg/L [209]. This evidence is consistent with the findings of the Smith group, where the metformin levels in all patients did not exceed the range according to their renal function [204]. In conclusion, suitable doses of metformin are not associated with unsafe plasma lactate and metformin levels in individuals with ACLD [204].

### 4.3. Monitoring Guidelines for Patients on Metformin Therapy

Metformin was first approved in the United States in 1995 [210] and has long been considered as a first-line therapy by the American Diabetes Association (ADA) and the European Association for the Study of Diabetes [211], as well as by the American Association of Clinical Endocrinologists and American College of Endocrinology (AACE/ACE) [212]. Many new drug classes have been approved in the last few years; despite the availability of these new agents, metformin continues to be the first-line agent for most patients with T2DM [24], with the notable exclusion of patients with heart failure. When compared with other glucose-lowering drugs, cardiovascular mortality was lower among metformin vs. sulfonylureas users, without any increase in body weight [213].

#### 4.3.1. Serum Creatinine Level Monitoring

Metformin was initially contraindicated for patients with renal disease due to concerns about lactic acidosis, and in the United States, it was recommended that metformin should be avoided in patients with a serum creatinine level > 1.4 mg/dL in women and >1.5 mg/dL in men [214].

Despite the restrictions, substantial evidence indicates that prescribers have continued to use metformin in patients with contraindications. The Diabetes Audit and Research in Tayside Scotland/Medicines Monitoring Unit (DARTS/MEMO) retrospectively tested the use of metformin in 691 patients with T2DM who developed contraindications, defined as two recordings of serum creatinine > 1.7 mg/dL on different days within 4 weeks. Only 10% discontinued therapy [214]. Despite the high number of patients treated with metformin in the presence of contraindications, only one episode of lactic acidosis occurred [214]. This patient was 72 years old, and the lactic acidosis was attributed to renal failure and acute myocardial infarction with massive myocardial damage [214].

Ekström et al. evaluated the Swedish National Diabetes Register to determine the safety and efficacy of metformin in patients with T2DM and varying levels of renal function [215]. Metformin had a lower incidence of any acidosis and severe infection in the range of the estimated glomerular filtration rates (eGFRs) of the 45 to <60 mL/min/1.73 m^2^ and eGFR ≥ 60 mL/min/1.73 m^2^ groups, with adjusted hazard ratio (HR) = 0.85 (95% CI = 0.74 to 0.97) and adjusted HR 0.91 (95% CI = 0.84 to 0.98), respectively [215]. Metformin was also associated with reducing all-cause mortality in the eGFR ≥ 60 mL/min/1.73 m^2^ group with an adjusted HR of 0.87 (95% CI = 0.81 to 0.94) [215]. These subgroup analyses did not reveal any increased risk of CVD, any acidosis, or severe infection, or all-cause mortality from metformin monotherapy [215].

A Cochrane review confirmed the incidence of lactic acidosis in those treated with metformin compared to those who were not. There were no fatal or non-fatal lactic acidosis cases in the metformin users group and the non-metformin group [215].

In the years between 2012 and 2015, in response to the evidence of continued use in renal insufficiency and its safety in doing so, as well as citizens’ petitions, the FDA decided to change the renal restrictions on the use of metformin in mild to moderate kidney disease [216]. They now recommend using eGFR instead of serum creatinine to determine if a patient with reduced renal function can safely take metformin. The new recommendation states that metformin is contraindicated in patients with an eGFR < 30 mL/min/1.73 m^2^ [217]. Metformin treatment should not be initiated in Those with an eGFR between 30 and 45 mL/min/1.73 m^2^ [217]. If the eGFR falls between 30 and 45 mL/min/1.73 m^2^ during metformin treatment, providers should assess the risks and benefits associated with continued use [217]. The AACE/ACE consensus statements on diabetes management agree with the FDA statement but recommend a reduced metformin dose in those with an eGFR between 30 and 45 mL/min/1.73 m^2^ [212].

In conclusion, new changes continue to occur regarding metformin use and monitoring. Prescribers should use eGFR cut-off points to determine the appropriateness of metformin therapy in patients with renal dysfunction. In patients with an eGFR of 30 to 45 mL/min/1.73 m^2^, the FDA recommends that metformin use should be continued with an increased frequency of monitoring of renal function. The available data indicate that metformin can reduce mortality, even in those with an eGFR between 30 and 60 mL/min/1.73 m^2^, and should not be used for those with an eGFR < 30 mL/min/1.73 m^2^ (Figure 2).

#### 4.3.2. Vitamin B_12_ Monitoring

Since the early 1970s, vitamin B_12_ deficiency has been reported with metformin use [218,219], with an incidence as high as 9.5% to 31% [218,220]. There is some concern about interpreting the incidence reported from these studies, as there is a wide variation in the literature regarding the definition of vitamin B_12_ deficiency. The exact mechanism is unknown, but it is theorised that metformin antagonises the calcium cation required in the ileal absorption of the vitamin [221]. In evaluating potential risk factors for metformin-induced vitamin B_12_ deficiency, the dose of metformin and its duration are two of the most significant risk factors for its development [221]. As some of the manifestations of vitamin B_12_ deficiency can occur as complications in diabetes, it is important to properly evaluate a patient treated with metformin for vitamin B_12_ deficiency. The neurologic manifestations of an untreated vitamin B_12_ deficiency can be irreversible, so identification and treatment (by oral or intramuscular route) are prudent [222]. A Cochrane review set out to determine the difference in efficacy of oral versus intramuscular vitamin B_12_; it concluded that the oral supplementation was as efficacious as the intramuscular route in achieving a hematologic response [223]. Prophylactic treatment was also suggested in patients at risk, particularly in patients undergoing gastric bypass surgery [224]. Serum levels of vitamin B_12_ should be obtained from those treated with metformin doses >1000 mg/day and those treated for an extended duration greater than 3 years (Figure 2).

## 5. Conclusion and Future Perspectives

Most studies on metformin use for the treatment of T2DM, impaired fasting glucose, and decreased glucose tolerance did not involve a liver biopsy, making it difficult to monitor the progression of MASLD/MASH accurately. Performing a liver biopsy before metformin therapy in patients with T2DM could provide valuable insights into their response to treatment, specifically their NAS scores. Research has shown that metformin affects various pathways involved in cancer development and progression, including AMP-activated protein kinase, mammalian target of rapamycin, and insulin-like growth factor signalling. Metformin also has indirect anti-tumour effects by reducing insulin and insulin-like growth factor 1 levels, which promote tumour growth and proliferation. It can also impact lipid metabolism and inflammation, which play a role in hepatocarcinogenesis. Therefore, it is worthwhile to examine the safety of metformin in patients already diagnosed with HCC and its potential to halt or regress its progression. Additionally, studying the combination of metformin with GLP1-receptor agonists (like liraglutide and semaglutide) commonly used in patients with T2DM and obesity with cirrhosis can determine if it reduces hepatic fibrosis and improves survival or slows disease progression.

In conclusion, metformin also may have a potential role in the management of cholestatic liver disease due to its antioxidant actions (particularly via glutathione peroxidase enzyme) in an animal model with bile duct ligation-induced liver injury [225]. In this context, it is also relevant to the possible action of metformin with coordinated p-cymene ligand incorporated into Ru-based organometallic anticancer agents, with the aim of enhancing the cytotoxic activity of the complex.

## Figures and Tables

**Figure 1 metabolites-14-00186-f001:**
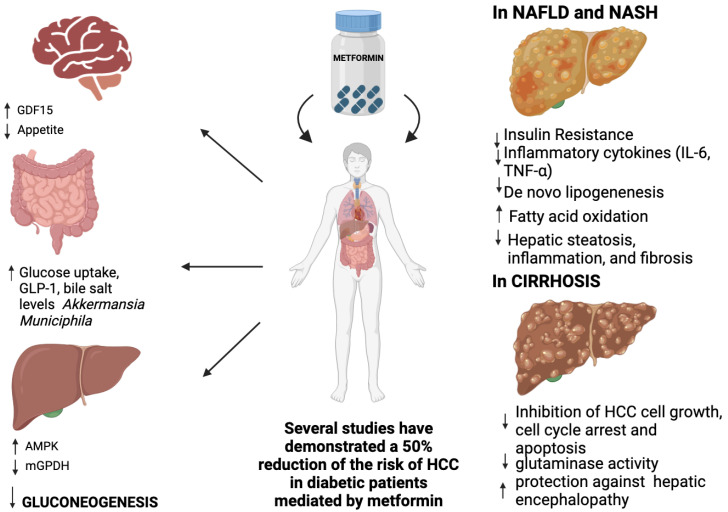
Mechanisms of action of Metformin.

**Figure 2 metabolites-14-00186-f002:**
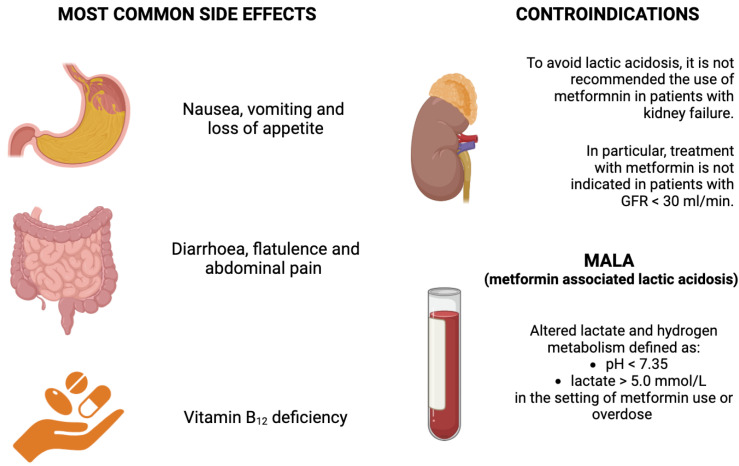
Most common side effects and contraindications of metformin and metformin-associated lactic acidosis (MALA).

## Data Availability

Data is available within the article.

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
