# Peer review of "Metformin and the Liver: Unlocking the Full Therapeutic Potential"

_metabolites, 2024, doi:10.3390/metabo14040186_

Round 1

Reviewer 1 Report

Comments and Suggestions for Authors

The authors' research focuses on a detailed examination of metformin, in conjunction with liver biopsy data. Metformin has been shown to have an impact on the development and progression of cancer, as well as exhibiting antitumor activity. Its involvement is multifaceted, encompassing the significant process of lipid metabolism. In conjunction with various receptors, it is possible to monitor the impact of metformin on the body. Nevertheless, the actual clinical application of metformin in different diseases in vivo necessitates caution due to limitations in relevant clinical trials, and the dosage ratio should be considered when developing novel metformin-based treatments. Among the benefits of the work, it is worth noting the competent and thorough utilization of literature sources. This undoubtedly underscores the significance of this research, which aids in comprehending the study of metformin and assessing its potential for future biochemical applications.

This article covers a comprehensive examination of the effects of metformin on the body. However, the authors should make some corrections to the comments before publishing the manuscript.

 1. The article should be supplemented with additional sections on metformin's effects on lipid metabolism, as this substance restores mitochondrial function

2. A separate section on liver fibrosis, including detailed examples and illustrations, should be included. 

3. Information should be provided about medical and chemical liver damage (DILI). 

4. Additional research in these areas would expand the potential use of metformin for treating various liver conditions. Some relevant references can be found in (10.2174/1381612824666181003114108 and 10.1039/C7RA06514K).

Author Response

REVIEWER#1

Comments to the Author

The authors' research focuses on a detailed examination of metformin, in conjunction with liver biopsy data. Metformin has been shown to have an impact on the development and progression of cancer, as well as exhibiting antitumor activity.  Its involvement is multifaceted, encompassing the significant process of lipid metabolism. In conjunction with various receptors, it is possible to monitor the impact of metformin on the body. 

Nevertheless, the actual clinical application of metformin in different diseases in vivo necessitates caution due to limitations in relevant clinical trials, and the dosage ratio should be considered when developing novel metformin-based treatments

. Among the benefits of the work, it is worth noting the competent and thorough utilization of literature sources. This undoubtedly underscores the significance of this research, which aids in comprehending the study of metformin and assessing its potential for future biochemical applications.

This article covers a comprehensive examination of the effects of metformin on the body. However, the authors should make some corrections to the comments before publishing the manuscript.

We would like to thank the reviewer for acknowledging the quality of our paper and for helping us further improve the manuscript before publication.

  • The article should be supplemented with additional sections on metformin's effects on lipid metabolism, as this substance restores mitochondrial function. 

Thank you very much. We added two separate paragraphs, 2.2.7 and 2.2.8, to deepen the role of metformin in mitochondrial homeostasis and lipid metabolism, respectively.

  • A separate section on liver fibrosis, including detailed examples and illustrations, should be included.

Thank you for your suggestion. However, we do not consider that “Liver fibrosis” needs a separate paragraph. We explore the subject exhaustively in paragraph 3.1.2, and in Figure 1, the principal mechanisms of action of metformin in NAFLD/MASLD, NASH/MASH, and Advanced Chronic Liver Disease are reported.

  • Information should be provided about medical and chemical liver damage (DILI).

We support the reviewer's suggestion and, thereby, added a specific paragraph (3.3) about the topics “Drug-induced liver disease” and “Chemical-induced liver injury.”

  • Additional research in these areas would expand the potential use of metformin for treating various liver conditions. Some relevant references can be found in (10.2174/1381612824666181003114108 and 10.1039/C7RA06514K).

Thank you very much. We mentioned the studies suggested in the paragraph “Conclusion” to highlight the potential role of metformin in more than metabolic diseases.

Reviewer 2 Report

Comments and Suggestions for Authors

Manuscript Number: metabolites-2927335

Journal Title: Metabolites (ISSN 2218-1989)

Title: Metformin and the Liver: Unlocking the Full Therapeutic Potential

Article Type: Review

Corresponding Authors: Federico Ravaioli

In this study, the authors have reviewed the Metformin effect against Liver diseases. The authors reviewed previous basic research papers and recent clinical work, citing important reports between basic and clinical studies. The review is considering a report arising from the diverse scientific backgrounds of researchers.

The authors are nicely reviewing, and the conclusions are justified and stated well based on many reports.

I recommend this paper for the “Metabolites” journal.

 This article does not satisfy the journal's publishing guidelines at this point. The authors should correct the formatting of the document and cite the original references (#35, 36, 39).

Author Response

REVIEWER#2

Comments to the Author

In this study, the authors have reviewed the Metformin effect against Liver diseases. The authors reviewed previous basic research papers and recent clinical work, citing important reports between basic and clinical studies. The review is considering a report arising from the diverse scientific backgrounds of researchers.

The authors are nicely reviewing, and the conclusions are justified and stated well based on many reports.

I recommend this paper for the “Metabolites” journal.

We would like to thank Reviewer 2 for appreciating our paper.

This article does not satisfy the journal's publishing guidelines at this point. The authors should correct the formatting of the document and cite the original references (#35, 36, 39).

Thank you for your feedback. We corrected the document's formatting and replaced the references you mentioned with the original ones.

Reviewer 3 Report

Comments and Suggestions for Authors

The review titled "Metformin and the Liver: Unlocking the Full Therapeutic Potential" by Perazza et. al., is a very comprehensive review of the drug metformin in contributing to liver diseases. The manuscript is very well written and is a detailed literature survey of current scientific knowledge in the field. I have only one minor suggestion: Please consider removing the"Materials and Methods" section. In this context it is unnecessary. Thank you.

Author Response

REVIEWER#3

Comments to the Author

The review titled "Metformin and the Liver: Unlocking the Full Therapeutic Potential" by Perazza et. al., is a very comprehensive review of the drug metformin in contributing to liver diseases. The manuscript is very well written and is a detailed literature survey of current scientific knowledge in the field.

Thank you for taking the time to provide us with your positive feedback on our manuscript.

I have only one minor suggestion: Please consider removing the "Materials and Methods" section. In this context it is unnecessary. Thank you.

We thank you for your suggestion. We agreed and removed the "Materials and Methods" section.
